# Role of the Uteroplacental Renin–Angiotensin System in Placental Development and Function, and Its Implication in the Preeclampsia Pathogenesis

**DOI:** 10.3390/biomedicines9101332

**Published:** 2021-09-27

**Authors:** Lucile Yart, Edith Roset Bahmanyar, Marie Cohen, Begoña Martinez de Tejada

**Affiliations:** 1Department of Pediatrics, Gynecology and Obstetrics, University Hospitals of Geneva, University of Geneva, 1211 Geneva, Switzerland; Lucile.Yart@unige.ch (L.Y.); Marie.Cohen@unige.ch (M.C.); 2Consultant in Obstetrics and Gynecology, 1000 Lausanne, Switzerland; edithroset@yahoo.fr

**Keywords:** angiotensin, placental development, uteroplacental renin–angiotensin system, preeclampsia

## Abstract

Placental development and function implicate important morphological and physiological adaptations to thereby ensure efficient maternal–fetal exchanges, as well as pregnancy-specific hormone secretion and immune modulation. Incorrect placental development can lead to severe pregnancy disorders, such as preeclampsia (PE), which endangers both the mother and the infant. The implication of the systemic renin–angiotensin system (RAS) in the pregnancy-related physiological changes is now well established. However, despite the fact that the local uteroplacental RAS has been described for several decades, its role in placental development and function seems to have been underestimated. In this review, we provide an overview of the multiple roles of the uteroplacental RAS in several cellular processes of placental development, its implication in the regulation of placental function during pregnancy, and the consequences of its dysregulation in PE pathogenesis.

## 1. Introduction

Successful pregnancy progression and achievement requires placental development that constitutes a unique interface for maternal–fetal exchanges. Within 6 days after fertilization, the outer layer of the blastocyst, differentiated into trophoblasts, starts to invade the maternal decidua to form the placenta [1,2]. Trophoblasts differentiate along either the villous or the extravillous pathways. Villous cytotrophoblastic cells fuse to give rise to the syncytiotrophoblast (STB) that supports specific functions, including hormone secretion, gases and nutrients exchanges, and the establishment of an immunological barrier [2,3]. Extravillous cytotrophoblasts (evCTB) invade the maternal uterus and are thus involved in the anchoring of chorionic villi in the uterus and the remodeling of the spiral arteries [2,3], ensuring placental vascularization.

The placental development that occurs during the first trimester of pregnancy under hypoxia promotes the formation of the STB [4] and angiogenesis [5]. Among other processes such as oxidative and endoplasmic reticulum stresses or autophagy [6,7], the renin–angiotensin system (RAS) plays a crucial role in placental development and functions [8]. The RAS is primarily known for its important systemic regulation of blood pressure, and fluid and electrolytes homeostasis, mainly mediated by angiotensin II (Ang II). Ang II preferentially binds to the Ang II type 1 receptor (AT1R), expressed in several organs including the kidneys, adrenal glands, heart, brain, and blood vessels [9]. This receptor has also been found in the placenta, in the STB, during the entire pregnancy [10]. The main functions of AT1R are vasoconstriction following activation in endothelial cells, and the induction of aldosterone secretion when activated in adrenal glands. Ang II also binds to Ang II type 2 receptor (AT2R), the activation of which mainly results in vasodilatation and anti-inflammatory effects [11]. Ang II is an octapeptide that is derived from angiotensinogen (Agt). The secreted Agt is first cleaved by renin to generate angiotensin I (Ang I), an inactive peptide (Figure 1). The kidneys are the primary source of renin, which is secreted in response to lowered blood pressure, but other tissues also locally synthesize this enzyme. Ang I is then mostly converted into Ang II by the angiotensin-converting enzyme (ACE). Alternatively, Ang I is metabolized by either the angiotensin-converting enzyme 2 (ACE2) or a membrane metalloendopeptidase (MME) to generate angiotensin (1–9) (Ang (1–9)) and angiotensin (1–7) (Ang (1–7)), respectively. Ang (1–7) can also be generated by the action of either the ACE on Ang (1–9) or the ACE2 directly on Ang II [12] (Figure 1). The importance of Ang (1–7) lies in its vasodilatory action, through the binding to the Mas receptor, which antagonizes the action of Ang II on AT1R [13]. Ang II can also be converted into angiotensin (2–8) (Ang III) by the aminopeptidase A, which principally binds AT2R and, to a lesser extent, AT1R to emphasize the effects of Ang II [14] (Figure 1). Finally, Ang III can undergo an additional cleavage by either the aminopeptidase B or the aminopeptidase N to generate angiotensin (3–8) (Ang IV), which binds to its specific receptor, Ang IV receptor (AT4R) [15] (Figure 1) to increase blood flow [16]. Within the placenta, AT4R activation by Ang IV in evCTB enhances the cell invasive capacities [10]. Changes in the systemic RAS regulation are essential for maternal hemodynamic adaptations to pregnancy but may also be implicated in the pathophysiology of several pregnancy disorders [17].

In addition to the systemic RAS, RAS components are synthesized in many organs such as the brain, heart, ovaries, uterus, and placenta, constituting local RAS [17,18]. In the early 1990s, many groups reported the presence of several RAS components in the maternal decidua, as well as in fetal–placental tissues. A high level of Ang II was detected in placental villi [19,20] and expression of pro-renin, ACE, and Agt was observed in both placental and fetal tissues [19,21,22]. The two Ang II receptors, AT1R and AT2R, have been detected in trophoblastic cells [23]. AT1R, which is the preferential binding site for Ang II, is expressed in high amounts throughout the pregnancy at both the transcript and protein levels [21,23,24,25]. AT2R was weakly detected, and only at the protein level in the human placenta [23], but its expression was shown to increase with pregnancy progression in rodent placenta [26]. Renin, Agt, ACE, and AT1R were also shown to be expressed in the maternal decidua [17,23,27,28], suggesting the existence of two local pregnancy-related RAS, a decidual and a placental one, together forming the uteroplacental RAS. An increasing number of studies have highlighted the significance of the local uteroplacental RAS in placental development (trophoblastic cells proliferation, migration, and invasion, and vascular remodeling) and functions (regulation of the uteroplacental blood flow, hormone secretion, nutrient transport, and immune modulation).

In this review, we will focus on the roles of the uteroplacental RAS in placental development and functions, and the pathogenesis of pregnancy disorders such as preeclampsia.

## 2. Roles of the Uteroplacental RAS

Ang II and other RAS components are well known for their effects on systemic vascular tone [29] and many other physiological processes, including angiogenesis [30,31], cell proliferation [31,32], and apoptosis [33,34], all of which take place in placental development and functions (Figure 2).

### 2.1. Placental Development

The implication of the uteroplacental RAS in pregnancy progression and placental development starts as early as week 2 of embryonic development by facilitating the implantation of the blastocyst [35,36]. Ang II promotes decidual cell differentiation and increases endometrial cell permeability [37], allowing the invasion of trophoblastic cells into the maternal endometrium. Several studies also highlighted the important roles of Ang II and AT1R signaling for the regulation of trophoblastic cell proliferation. In vitro experiments conducted on first-trimester placental explants showed an increase in evCTB proliferation in response to Ang II, as demonstrated by increased numbers of Ki67- and BrdU-positive cells [23,38]. This effect was abolished following the pharmacological blockade of either AT1R alone or both AT1R and AT2R, demonstrating that Ang II-enhanced evCTB proliferation mainly involved AT1R signaling and probably AT2R to a lesser extent. However, the effect of AT2R blockade alone was not mentioned by the authors [38]. The treatment of first-trimester human placental explants with Ang II enhanced the transcription of the plasminogen activator inhibitor-1 (PAI-1) gene through AT1R signaling [38], which promoted hypercoagulation. Increased levels in PAI-1 in response to Ang II were also measured by Xia et al. [39] in the extravillous trophoblastic cell line HTR-8, and this was associated with an inhibition of the cell invasive capacity. However, the role of Ang II in the regulation of trophoblastic cell migration and invasion is controversial. Ishimatu et al. [40] observed increased invasive capacities in the choriocarcinoma cell line BeWo in response to Ang II. This observation was further reinforced by the studies conducted by Hering et al. [41] and Williams et al. [10], who reported enhanced invasion and motility in the first-trimester human placental explants treated with Ang II.

Despite the multiple functions of the uteroplacental RAS in placental development, its implications in spiral arteries remodeling and angiogenesis are most commonly studied. All the RAS components are expressed in and around human spiral arteries [27]. Interestingly, the upregulation of several RAS components in the maternal decidua, as well as in the endothelial and vascular smooth muscle cells of the spiral arteries and the perivascular stromal cells during the first trimester of pregnancy, coincides with important vasoactivity of the spiral arteries, suggesting a major role in pregnancy-induced vascular remodeling [27]. More recently, Ang II has been shown to regulate the expression of the two soluble angiogenic factors, vascular endothelial growth factor (VEGF) and placental growth factor (PlGF), through the activation of AT1R in vascular endothelial and smooth muscle cells [15,42]. VEGF and PlGF then promote the proliferation of both cell types [42]. In normal pregnancies, VEGF and PlGF, also expressed in the trophoblasts and the maternal decidua, play crucial roles in placental vascular development by regulating the vascularization within the chorionic villi [43,44]. Once secreted, these two factors promote endothelial cell proliferation and vascular expansion [42,43,44]. In humans, placental vascularization continuously develops, forming new capillaries from week 3 to week 26 of gestation [43]. This occurs in a hypoxic condition that is known to be crucial in early pregnancy for trophoblast invasion and spiral artery remodeling and regulates the expression of VEGF and its receptor, Fms-like tyrosine kinase 1 (Flt1), through the activation of AT1R and AT2R expression within the uteroplacental unit [8]. AT1R signaling in trophoblast cells also promotes the expression of the anti-angiogenic factors soluble Flt1 (sFlt1) and soluble endoglin (sEng) [45], thus balancing placental angiogenesis. This soluble VEGF receptor, sFlt1, expressed in the trophoblasts, is a spliced form of Flt1 with a preserved extracellular domain and truncated transmembrane and intracellular signaling domains. Thus, the concurrent binding of VEGF and PlGF to the circulating sFlt1 reduces the interaction of the two angiogenic factors with their cell-surface receptors. In normal pregnancies, the circulating level of sFlt1 is low in early pregnancy, during placental vascular development, and increases near term, during the last 2 months of pregnancy, to prepare for delivery by promoting vasoconstriction and activation of the coagulation cascade [43]. The second anti-angiogenic factor promoted by AT1R signaling in trophoblastic cells, sEng, is a truncated form of endoglin, a cell-surface receptor for the transforming growth factor-β (TGF-β) [44,46]. Circulating sEng binds and antagonizes TGF-β, thus inhibiting the synthesis of the vasodilator molecule nitric oxide. Similar to VEGF and PlGF, these two anti-angiogenic factors require a fine regulation of the uteroplacental RAS. In addition to being upregulated by the AT1R activation in trophoblasts [47,48,49,50], sFlt1 expression is balanced through the activation of AT2R and the Mas receptor by the binding of Ang II and Ang (1–7), respectively [13,15]. This is not the case for sEng [47], suggesting that sEng regulation essentially involves AT1R signaling and does not require AT2R or the Mas receptor.

### 2.2. Placental Function

The uteroplacental RAS has been especially shown to regulate the uteroplacental blood flow as well as the endocrine function of the STB and participates in nutrient transport and immune modulation.

The regulation of the placental blood flow by Ang II seems to involve complex mechanisms [36]. Ang II has been shown to induce ex vivo dose-dependent vasoconstriction in chorionic plate arteries collected from term pregnant women [51], while Svane et al. [52] reported a limited effect of both Ang I and Ang II on fetal villous arteries contractility, but a significant increase in intra-myometrial arteries’ contractility in response to both peptides [52]. This differential effect could be due to spatial and temporal changes in AT1R and AT2R expression. Experiments conducted on rodent models showed that AT2R is specifically expressed in the vascular smooth muscle cells of the umbilical cord and placenta [53], upregulated in uterine arteries during pregnancy in response to estrogens, and rapidly decreased after parturition [26]. The increase in AT2R expression in uterine arteries positively correlates with the increase in uterine blood flow [26], and pregnant mice defective for AT2R displayed a progressive increase in blood pressure during pregnancy [53], suggesting that AT2R signaling in uteroplacental blood vessels is important to balance placental perfusion. On the contrary, AT1R expression seems to remain high and steady in chorionic plates as well as in placenta and umbilical cord blood vessels during the entire pregnancy [26,53]. This receptor would primarily serve for the preferential constriction of the fetal peripheral circulation [54].

Another major role of the uteroplacental RAS during pregnancy is the regulation of the endocrine function supported by the STB. The STB produces and secretes several hormones essential for pregnancy progression, including steroid hormones (estrogens and progesterone), placental lactogen hormone (hPL), and pregnancy-specific β_1_ glycoprotein (SP1). Remarkably, Ang II has been shown to influence this endocrine activity. In vitro experiments conducted on placental explants highlighted increased hPL and SP1 secretions [55,56]. Ang II also enhanced the production of estrogens by promoting the aromatization of androgens [57]. The use of specific pharmacological blockers demonstrated that this function was mediated by AT1R [55,57] and involves calcium signaling pathways [56].

Regarding the role of the uteroplacental RAS in the regulation of nutrient transport, several factors are known to influence nutrient transport across the placenta, including blood flow, exchange surface, metabolic activity, and expression and activity of transporter proteins [58]. Among these transporter systems, System A facilitates the transfer of small, non-essential amino acids. In 2006, Shibata et al. [59] reported that Ang II-mediated activation of AT1R in primary villous fragments decreased the System A activity [59] and thereby affected amino acid transport across the placenta.

Additionally, the uteroplacental RAS may be involved in placental immune modulation. A recent study showed that Ang II affected in vitro monocyte adhesion when co-cultured with trophoblastic cells. This was mediated by a transiently increased expression of the CX3CL1 chemokine in response to AT1R activation [60].

## 3. Regulation of the Uteroplacental RAS and Associated Placenta Dysfunction

### 3.1. Physiological Regulation

Considering the pivotal role of the uteroplacental RAS in both placental development and function, its regulation appears to be a key process for successful progression of pregnancy. The uteroplacental expression of various RAS components, including renin, Agt, ACE2, AT1R, and the prorenin receptor, is higher in early pregnancy and then decreases within the second trimester [22]. As mentioned above, early placental development occurs in hypoxic conditions that have been shown to enhance the expression of AT1R in trophoblastic cells [61,62], the activation of which promotes the production of pro-angiogenic factors [61,63] required for placental vascular development (Figure 3).

Interestingly, the regulation of the uteroplacental RAS by hypoxia seems to involve microRNAs (miRNA). Several miRNAs known to target the expression of RAS components are regulated by oxygen levels [64,65]. Placental miRNAs, which are low in early pregnancy, are upregulated with pregnancy progression, and some miRNAs may serve to decrease the expression of RAS components in trophoblastic cells [66] (Figure 3). However, despite down-regulation of renin by miRNAs in trophoblastic [66], the renin level is maintained by the human chorionic gonadotropin (hCG) secreted by the differentiated STB, as hCG has been shown to stimulate the expression of the gene encoding for renin and possibly Agt [67] (Figure 3).

In late pregnancy, RAS components, especially, renin, ACE, and Agt, were shown to be upregulated in the maternal decidua and myometrium around parturition [22,68]. ACE and Agt transcript levels increased in laboring myometrium, compared to pregnant myometrium collected before labor onset [68], suggesting an involvement of the RAS in myometrial contractions. Furthermore, other RAS components, including renin, prorenin receptor, Agt, and Mas receptor, increased in maternal decidua before labor onset and were higher in women carrying female fetuses than male fetuses [69]. Decidual level of renin decreased after parturition only in cases of female fetuses [69], confirming that the regulation of the uteroplacental RAS is influenced by the fetal sex (Figure 3), probably through the passage of fetal cells to maternal circulation [69].

### 3.2. Uteroplacental RAS Dysregulation

The regulation of uteroplacental RAS appears to be a key process for correct placental development and pregnancy progression. RAS dysregulation, especially in early pregnancy, would lead to incorrect trophoblastic cell differentiation as well as pro- and anti-angiogenic factor imbalance, and therefore cause defects in placental perfusion and pregnancy disorders.

Reduced placental perfusion is associated with severe pregnancy disorders such as fetal growth restriction (FGR) or preeclampsia (PE, see below). Delforce et al. [70] showed that AT1R and ACE2 were downregulated in the placenta with FGR. ACE2 is one of the enzymes that converts the vasoconstrictor agent Ang II into Ang (1–7), which has a vasodilatory effect. Therefore, they concluded that a reduction in ACE2 would result in a low level of Ang (1–7) and the accumulation of Ang II, which would further lead to important vasoconstriction and reduced perfusion of the placenta [70].

In addition to the regulation of Ang II precursors and derivates, the expression level of the different receptors is also important for placental perfusion. As mentioned above, during early pregnancy the placental expression of AT1R and AT2R is enhanced by hypoxia, which plays a crucial role in placental perfusion, especially by regulating trophoblast invasion and spiral artery remodeling [62]. However, extended hypoxia beyond the first trimester of pregnancy, which is known to alter placental function [62], also enhances both AT1R and AT2R expression within the uteroplacental unit [8]. Overactivation of AT1R in trophoblast cells has been shown to induce important increases in circulating levels of the anti-angiogenic factors sFlt1 and sEng [45], PAI-1, and tissue factor [71]. In addition to promoting hypercoagulation and anti-angiogenic factors, prolonged hypoxia inhibits the expression of PlGF in trophoblasts [43]. The high circulating level of sFlt1, resulting from AT1R activation, which concurrently binds VEGF and PlGF, together with the decrease in PlGF secretion, affects trophoblast invasion and spiral artery remodeling, thereby reinforcing defects in placental perfusion. Incorrect placental perfusion then reinforces the hypoxia, thus maintaining a negative loop. sEng has been shown to follow a similar pattern to sFlt1, involving AT1R signaling [49,72]. This contributes to the amplification of the vascular damages mediated by sFlt [12,46]. Activation of AT1R also promotes oxidative stress, by stimulating Nicotinamide Adenine Dinucleotide Phosphate Hydrogen (NADPH)-oxidase, and inflammation, mediated by nuclear factor-κB (NF-κB) [73].

## 4. Involvement of the Uteroplacental RAS in Preeclampsia Pathogenesis

PE is a pregnancy-specific syndrome commonly characterized by new-onset hypertension and proteinuria or symptoms of systemic disease [44,74] occurring after 20 weeks of gestation. The diagnosis of PE syndrome varies depending on the signs and symptoms considered in the definition. As a consequence, combined with unequal access to hospital care, the incidence varies from 2 to 10% of pregnancies [75,76,77,78] depending upon the country. Despite this relatively low incidence, pregnancies complicated by PE represent about 30% of maternal mortality, mainly in developing countries, with an estimate of 63,000 maternal deaths per year [78]. PE is associated with a poor prognosis for both mother and infant [76,79,80]. In addition to short-term complications, PE has now been recognized to be associated with long-term morbidities such as hypertension, ischemic heart disease, or end-stage renal disease [44,79] in about 20% of affected women [46].

The manifestation of the PE symptoms before or after 34 weeks of pregnancy categorizes this disease as early- or late-onset PE, respectively. Late-onset PE is believed to be a “maternal syndrome” that would be due to maternal genetic predispositions to cardiovascular and metabolic disorders, in association with the physiological placental senescence. However, early-onset PE is considered more of a “placental syndrome”, arising from a defect in placental development (see [78] for review). Despite convergent symptoms, the different etiologies for early- and late-onset PE make it two distinct disorders. In this part of the review, we will focus on early-onset PE, which has more dramatic consequences for women and children.

The understanding of the primary pathogenesis of PE is still incomplete. However, there is now substantial evidence that this pregnancy disorder is supported by reduced uteroplacental blood flow, leading to placental ischemia [81,82]. It is associated with a limited systemic RAS upregulation compared to normotensive pregnant women [17,45,74], notably shown by lower renin, Ang I, Ang II, and Ang (1–7) circulating levels [74]. Unexpectedly, despite a reduced RAS upregulation, PE women develop hypersensitivity to Ang II within the first 10 weeks of gestation [83] (i.e., several weeks before the onset of clinical symptoms [44]) such as systemic hypertension.

In addition to the indisputable involvement of the systemic RAS in PE pathogenesis, the placenta seems to play a major role in this pregnancy disorder. Women with hydatidiform mole (a trophoblastic cell tumor without fetus) frequently develop PE [46], demonstrating the accessory role of the fetus in PE pathogenesis and suggesting that the underlying mechanisms of PE pathogenesis are related to placental development [84].

In some cases, PE is associated with genetic polymorphisms of the transcription factor storkheadbox 1 (Stox1) [85]. In vitro, the mutation of the *Stox1* gene in trophoblastic cells appears to induce trophoblast dysfunction, including altered syncytialization, membrane repair, and redox equilibrium [86]. A recent study showed that *Stox1* knock-out mice developed gestational hypertension associated with an upregulation of the uteroplacental RAS [87]. This was demonstrated by increases in renin placental gene expression and protein levels, and by the rescue of the phenotype after treatment with an Ang II receptor blocker [87]. However, independently of *Stox1* polymorphism, women with PE have a differentially regulated uteroplacental RAS compared to normotensive pregnant women. Whereas AT1R expression levels in placentas seems to be high [25] with similar tissue localization in both PE and normotensive pregnant women [88], only PE patients showed upregulated AT1R expression in the maternal decidua [17,45] (Figure 4). Ex vivo chorionic plate arteries isolated from normotensive pregnant women displayed increased contractility when treated with culture medium from PE placental explants compared with culture medium from normotensive placental explants [89]. This suggests that the PE placenta secretes factors that induce vasoconstriction of placental arteries. The addition of a specific AT1R antagonist inhibited the vasoconstrictor effect, while the AT2R blockade did not affect vasoconstriction [89], highlighting the key role of uteroplacental AT1R in PE development.

Additionally, AT1R activation has been shown to upregulate sFlt1 expression [49,71,90], primarily in trophoblastic cells [90] (Figure 4). Dysregulation in anti-angiogenic factor production that occurs in PE women and leads to increased sFlt1 circulating levels has been shown to play an important role in PE pathogenesis [18,28,43,44,71]. Clinically, the risk for a pregnant woman with hypertension to develop PE is assessed by measuring the ratio of circulating sFlt1/PlGF [91,92]. The increase in sFlt1 in PE women is positively correlated with the severity of the disease [50]. The highest sFlt1 circulating levels measured in PE women compared with normotensive pregnant women resulted in a reduction in VEGF and PlGF bioavailability [28,43,44,46,93]. The decrease in free VEGF levels induces the expression of endothelin-1, which further contributes to reduce the secretion of renin and aldosterone. Low renin and aldosterone production in PE women reinforces the negative regulation of blood volume, thereby exacerbating poor placental perfusion [94].

## 5. Conclusions

The uteroplacental RAS, while not essential for pregnancy progression, appears rather to regulate multiple processes in placental development and function. More specifically, the activation of the uteroplacental RAS in early pregnancy is important for syncytialization and spiral artery remodeling, therefore ensuring efficient maternal–fetal exchanges. Modulations of the uteroplacental RAS that occurs throughout the pregnancy tightly balance the uteroplacental blood flow, nutrient transport, and hormone secretion. Dysregulation of the uteroplacental RAS results in altered placental development and function, leading to pregnancy disorders such as PE. In PE patients, the consequences of uteroplacental RAS dysfunction are not only local but have systemic effects on maternal organs and could lead to long-term morbidities. Due to the central role of the uteroplacental RAS in pregnancy, it should be more thoroughly investigated for the development of new therapies and diagnostic tools for RAS-associated pregnancy disorders.

In addition, the recent discovery of ACE2 as a receptor for the severe acute respiratory syndrome coronavirus 2 (SARS-CoV-2) further reinforces interest in this protein in the uteroplacental unit. ACE2 is expressed in both maternal decidua and placenta and could be involved in a possible vertical transmission of coronavirus disease (COVID)-19 from the mother to the fetus, as well as in COVID-19-associated miscarriages and stillbirths [95].

## Figures and Tables

**Figure 1 biomedicines-09-01332-f001:**
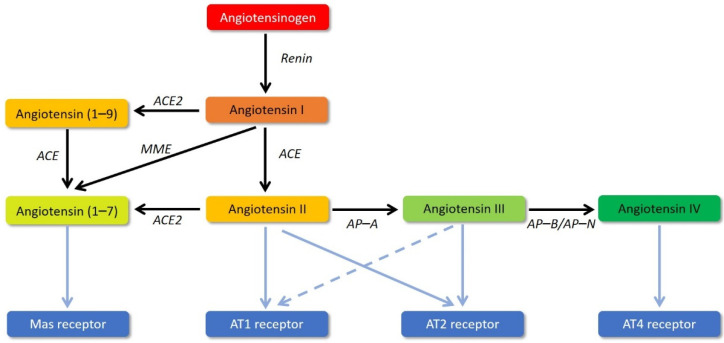
The renin–angiotensin system (RAS). The main effectors of the RAS, angiotensin II, angiotensin (1–7), angiotensin III, and angiotensin IV, all derivate from angiotensinogen. Each color corresponds to a conversion level. The different receptors (Mas, AT1, AT2, and AT4) are presented in blue boxes. Relations with their respective ligands are shown by blue arrows. The enzymes allowing the cleavage of the different peptides (black arrows) are written in black italics. ACE: angiotensin-converting enzyme; ACE2: angiotensin-converting enzyme 2; MME: membrane metalloendopeptidase; AP-A: aminopeptidase A; AP-B: aminopeptidase B; AP-N: aminopeptidase N.

**Figure 2 biomedicines-09-01332-f002:**
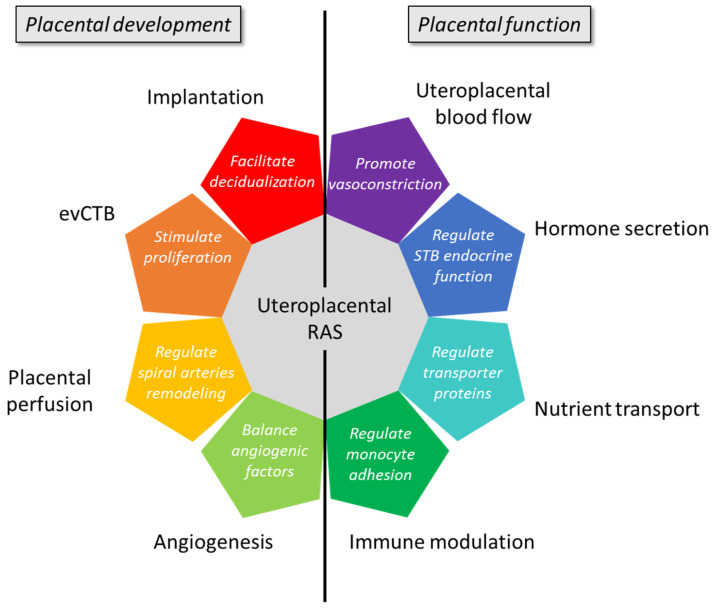
Role of the uteroplacental renin–angiotensin system (RAS) in placental development and function. evCTB: extravillous cytotrophoblasts; STB: syncytiotrophoblast.

**Figure 3 biomedicines-09-01332-f003:**
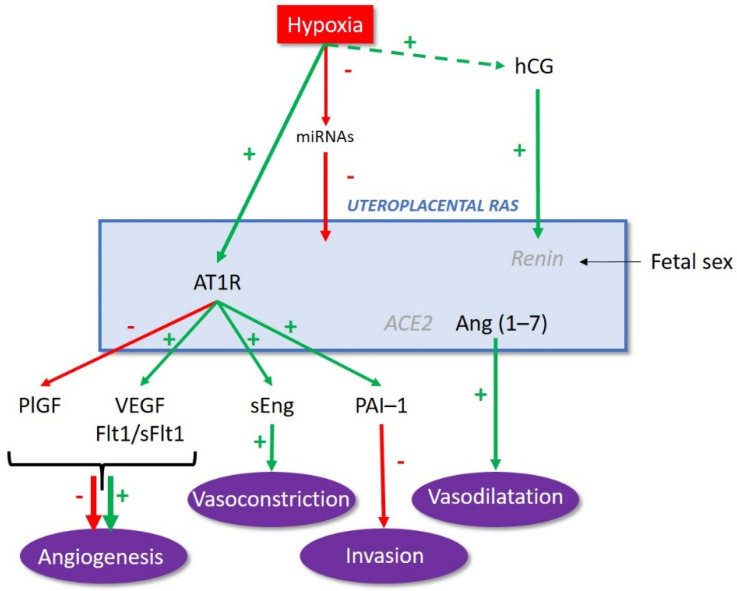
Regulation of the uteroplacental renin–angiotensin system (RAS). hCG: human chorionic gonadotropin; miRNA: microRNA; AT1R: angiotensin II type 1 receptor; ACE2: angiotensin-converting enzyme 2; Ang (1–7): angiotensin (1–7); PlGF: placental growth factor; VEGF: vascular endothelial growth factor; Flt1: Fms-like tyrosine kinase 1; sFlt1: soluble Fms-like tyrosine kinase 1; sEng: soluble endoglin; PAI-1: plasminogen activator inhibitor 1.

**Figure 4 biomedicines-09-01332-f004:**
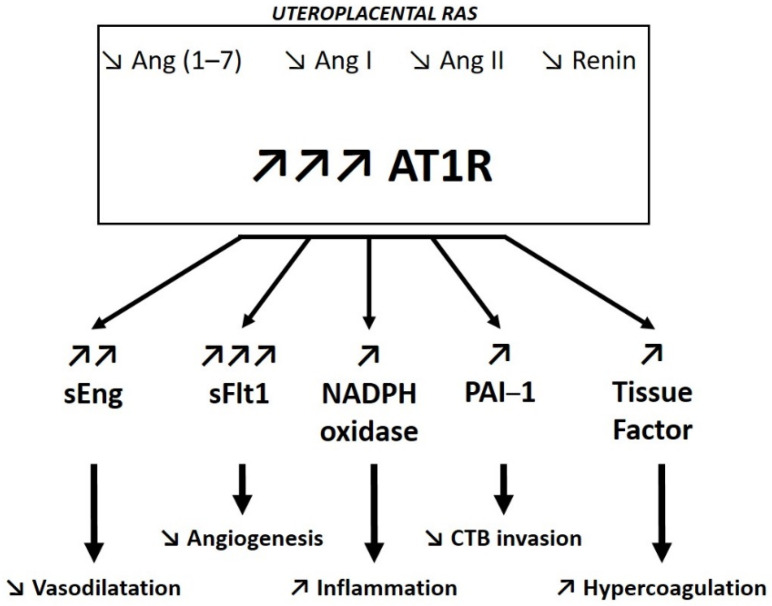
Uteroplacental renin–angiotensin system (RAS) dysfunction in preeclamptic women compared to normotensive pregnant women. Ang (1–7): angiotensin (1–7); Ang I: angiotensin I; Ang II: angiotensin II; AT1R: angiotensin II type 1 receptor; sEng: soluble endoglin; sFlt1: soluble Fms-like tyrosine kinase 1; NADPH: Nicotinamide Adenine Dinucleotide Phosphate Hydrogen; PAI-1: plasminogen activator inhibitor 1; CTB: cytotrophoblasts.

## Data Availability

Not applicable.

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
