# Peer review of "Role of the Uteroplacental Renin–Angiotensin System in Placental Development and Function, and Its Implication in the Preeclampsia Pathogenesis"

_biomedicines, 2021, doi:10.3390/biomedicines9101332_

Round 1
Reviewer 1 Report
This is a nice review which aimed at providing an overview of the multiple roles of the uteroplacental renin-angiotensin system in cellular processes of placental development, and its implication in the regulation of placental function as well as the consequences of its dysregulation in preeclampsia pathogenesis. The accumulated data is presented in a very intelligible fashion, the figures are very informative, and the reading is entertaining and interesting. I have not found anything major to revise and suggest the manuscript to be accepted in its form.
Author Response
We thank the reviewer for considering our manuscript, and for this nice comment.
Reviewer 2 Report
Thank you very much for allowing me to review this manuscript. in my opinion the Authors raise a very important topic and relevant issue. Unfortunately I cannot recommend this paper for publication in the present form. The paper need a major improvement.
A few specific concerns:
Tha manuscript needs clarification.
Some data and particularly references are given chaotically. Tha Authors chaotically refer to the cited data and citerd references.
For example on page 7 vers 248 Authors refer to the paper by Delforce and provide the reference number 70 . But paper by Delforce is a number 62 in the referenmces list. And at number 70 in the references list is the paper by Herse and laMarca.
Moreover it seems that quoted quotation in the text does not apply to this manuscript by Delforce.
In the paragraph Conclusions Authors refer to papaer No 95, while the list of the references gives references from 1 to 93. Similarly Authors refer to the reference number 94, which is also missing from the list of references in this paper (cited on page 8 vers 337).
Author Response
We thank the reviewer for considering our manuscript and for the helpful comments about the lack of clarity of the citations.
In the revised version of the manuscript, we have updated the references list. In addition to the automatic generation of the references list with a bibliographic software, all the citations have been verified manually.